# A Fatal A/H5N1 Avian Influenza Virus Infection in a Cat in Poland

**DOI:** 10.3390/microorganisms11092263

**Published:** 2023-09-09

**Authors:** Olga Szaluś-Jordanow, Anna Golke, Tomasz Dzieciątkowski, Dorota Chrobak-Chmiel, Magdalena Rzewuska, Michał Czopowicz, Rafał Sapierzyński, Michał Kardas, Kinga Biernacka, Marcin Mickiewicz, Agata Moroz-Fik, Andrzej Łobaczewski, Ilona Stefańska, Ewelina Kwiecień, Iwona Markowska-Daniel, Tadeusz Frymus

**Affiliations:** 1Department of Small Animal Diseases with Clinic, Institute of Veterinary Medicine, Warsaw University of Life Sciences-SGGW, Nowoursynowska 159c, 02-776 Warsaw, Poland; tadeusz_frymus@sggw.edu.pl; 2Department of Preclinical Sciences, Institute of Veterinary Medicine, Warsaw University of Life Sciences-SGGW, Ciszewskiego 8, 02-786 Warsaw, Poland; anna_golke@sggw.edu.pl (A.G.); dorota_chrobak@sggw.edu.pl (D.C.-C.); magdalena_rzewuska@sggw.edu.pl (M.R.); ilona_stefanska@sggw.edu.pl (I.S.); ewelina_kwiecien@sggw.edu.pl (E.K.); 3Chair and Department of Medical Microbiology, Medical University of Warsaw, Chałubińskiego 5, 02-004 Warsaw, Poland; tdzieciatkowski@wum.edu.pl; 4Division of Veterinary Epidemiology and Economics, Institute of Veterinary Medicine, Warsaw University of Life Sciences-SGGW, Nowoursynowska 159c, 02-776 Warsaw, Poland; michal_czopowicz@sggw.edu.pl (M.C.); kinga_biernacka@sggw.edu.pl (K.B.); marcin_mickiewicz@sggw.edu.pl (M.M.); agata_moroz_fik@sggw.edu.pl (A.M.-F.); iwona_markowska_daniel@sggw.edu.pl (I.M.-D.); 5Department of Pathology and Veterinary Diagnostics, Institute of Veterinary Medicine, Warsaw University of Life Sciences-SGGW, Nowoursynowska 159c, 02-776 Warsaw, Poland; rafal_sapierzynski@sggw.edu.pl; 6Veterinary Clinic Auxilium, Arkadiusz Olkowski, Królewska Str. 64, 05-822 Milanówek, Poland; mkardas94@gmail.com (M.K.); alobaczewski007@gmail.com (A.Ł.)

**Keywords:** cats, highly pathogenic avian influenza virus (HPAI), H5N1

## Abstract

A European Shorthair male cat, neutered, approximately 6 years of age, was presented to the veterinary clinic due to apathy and anorexia. The cat lived mostly outdoors and was fed raw chicken meat. After 3 days of diagnostic procedures and symptomatic treatment, respiratory distress and neurological signs developed and progressed into epileptic seizures, followed by respiratory and cardiac arrest within the next 3 days. Post-mortem examination revealed necrotic lesions in the liver, lungs, and intestines. Notably, the brain displayed perivascular infiltration of lymphocytes and histiocytes. Few foci of neuronal necrosis in the brain were also confirmed. Microscopic examination of the remaining internal organs was unremarkable. The A/H5N1 virus infection was confirmed using a one-step real-time reverse transcription polymerase chain reaction (RT-qPCR). The disease caused severe neurological and respiratory signs, evidence of consolidations and the presence of numerous B lines, which were detected on lung ultrasound examination; the postmortem findings and detection of A/H5N1 viral RNA in multiple tissues indicated a generalized A/H5N1 virus infection. Moreover, a multidrug-resistant strain of *Enterococcus faecium* was isolated in pure culture from several internal organs. The source of infection could be exposure to infected birds or their excrements, as well as contaminated raw poultry meat but, in this case, the source of infection could not be identified.

## 1. Introduction

Until a few decades ago, influenza was scarcely diagnosed in cats. However, increasing evidence indicates that several mammalian species, including cats, can be involved in the global circulation of type A influenza viruses (IAVs) [1]. Most feline influenza virus infections recorded to date were caused by low pathogenic strains that typically induce only subclinical infections or a mild respiratory disease [2]. These pathogens were generally transmitted to cats from humans [3,4], birds [1,5] and dogs [1,6,7]. However, due to high genetic variability, IAVs can rapidly alter their antigens or their ability to replicate in new host species [8]. Furthermore, they can also modify their virulence, typically leading in birds to the emergence of the highly pathogenic avian influenza (HPAI) [9]. These viruses can cause an acute, severe, and generalized disease with high mortality in wild and/or domestic birds [10]. In the past, such epidemics of “fowl plague” tended to fade away naturally, and each HPAI virus (HPAIV) eventually disappeared [9]. However, the HPAIVs carrying a H5N1 combination of surface proteins hemagglutinin (H) and neuraminidase (N), first identified in China in 1996, proved to be a game-changer, becoming endemic in Southeast Asia [11].

Since 2003, the A/H5N1 virus has caused countless outbreaks of HPAI, initially devastating poultry farming in Southeast Asia, and then spreading to Europe, Africa, and eventually to the Americas [12]. During worldwide expansion, new reassortments of this virus emerged by substituting N1 with other N subtypes (H5Nx viruses). The hemagglutinin gene of the H5Nx viruses that have been circulating for the past 20 years is classified into various phylogenetic clades [13]. The most widely circulating A/H5N1 viruses, of the 2.3.4.4b clade, have been detected in wild birds and domestic poultry in Europe, Africa, Asia, and North America [14]. In the last season (2021–2022), in Europe, this clade caused the largest epidemic in the history, involving 37 countries with a total of 2520 HPAI outbreaks in poultry [15]. In total, 50 million birds were killed by the virus or culled in the affected farms. The virus has been also detected in captive (227 cases), and wild birds (3867 cases). The epidemic re-emerged in Europe in 2023, also involving Poland, where 68 outbreaks in poultry (the last one 1 July) and 138 in wild birds (the last 14 July) were confirmed (Chief Vet. Off., https://www.wetgiw.gov.pl/nadzor-weterynaryjny/hpai, accessed on 27 August 2023) before the country was declared HPAI free on 2 August (https://www.woah.org/app/uploads/2023/08/poland-hpai-selfd.pdf, accessed on 27 August 2023). In Europe, between 3 December 2022 and 23 June 2023, there was a notable presence of HPAI clade 2.3.4.4b. outbreaks in domestic (726) and wild (2382) birds across 25 countries. It appears that there has been a decline in the frequency of disease outbreaks; however, a full comparison is not currently possible due to the ongoing 2022/2023 season [16,17,18].

Very rarely, the A/H5N1 virus can infect mammals, including domestic or wild cats [19,20,21]. Domestic cats, experimentally infected with this agent, developed fatal disease with pneumonia, neurological signs, and damage to internal organs [19,21]. However, although the A/H5N1 virus has been constantly present in wild and domestic bird populations of Southeast Asia, Europe, and Africa for over 20 years, cases of HPAI in cats were very rare. The first two outbreaks were recognized in tigers and leopards that had been fed infected poultry carcasses in Thai zoos [20,22]. Subsequently, sporadic cases have been documented in domestic cats in Thailand [23], Iraq [24], Germany [25], France [26], and recently in the USA [12] (https://bnonews.com/index.php/2023/04/another-cat-in-the-u-s-dies-of-h5n1-bird-flu/, accessed on 27 August 2023). In addition, subclinical A/H5N1 infections were identified in three cats in a shelter in Austria [27]. Other H5Nx viruses, such as A/H5N8, can also be pathogenic for mammals. Cases of fatal outcome have been described in foxes and seals [28]. Moreover, the A/H5N6 virus, which is believed to have originated in Sichuan province around the beginning of 2014, can rarely be pathogenic also for felids. By 2015, the first documented cases of A/H5N6 infections were observed in both a domestic cat and wild birds. By December 2016, cats in South Korea exhibited infections with A/H5N6, showing systemic pathological lesions [29,30].

In contrast to these sporadic feline cases, in June 2023, veterinary practitioners in various regions of Poland concurrently reported a number of cases of an acute, severe and highly fatal pneumonic disease with neurological signs in cats. At the time of writing, the A/H5N1 virus infection has been confirmed in 33 domestic cats and one captive caracal in Poland (Chief Vet. Off., https://www.wetgiw.gov.pl/main/komunikaty/Komunikat-VII-GLW-w-sprawie-choroby-kotow/idn:2302, accessed on 27 August 2023). This countrywide phenomenon, involving both outdoor and indoor cats, was exacerbated by rapid disease progression, limited diagnostic access, lack of established management guidelines and, finally, by a media-fueled panic.

This case report describes the clinical course, diagnostic methods, including lung ultrasound examination, and results of laboratory tests of a cat with A/H5N1 influenza virus infection from central Poland.

## 2. Materials and Methods

### 2.1. Patient Description

A European Shorthair male cat, neutered, around 6 years of age, weighing 5 kg, was presented to the veterinary clinic on 16 June 2023, due to apathy and loss of appetite. The patient had outdoor access and its diet included raw chicken meat. The cat was under the care of a private owner. The patient was regularly vaccinated against feline calicivirus, feline herpesvirus, panleukopenia, and rabies, according to the common vaccination schedule.

### 2.2. Blood Analysis

Venous blood was collected into one serum clot tube and one tube with EDTA, and sent to the veterinary laboratory for hematological and biochemical analyses.

### 2.3. Ultrasound Examination and X-ray

Imaging diagnostics of the thorax was performed in a short intravenous sedation with a combination of dexmedetomidine (Dexdomitor, 5 µg/kg b.w. i.v., Zoetis, Parsippany, New Jersey, USA) and butorphanol (Torphadine, 0.2 mg/kg b.w. i.v., Dechra, Northwich, Cheshire, UK). Ultrasound examination of the lungs and heart was performed using a Versana Premier, GE Medival System (Wuxi, Jiangsu, China) with a 12L-RS linear probe (frequency 10 MHz) and a microconvex 8C-RS ultrasound probe (8 MHz for the heart). X-ray examination of the chest was performed in two projections using an Ecoray-1060HF Portable X-ray Generator (Ecoray, Seoul, Republic of Korea).

### 2.4. Pathological Examination

The anatomopathological examination was performed on a fresh carcass. The necropsy was conducted according to standard procedures. Tissue fragments from the brain, lung, heart, liver, spleen, pancreas, kidney, mesenteric lymph node, and intestine were collected and fixed in a phosphate buffered 4% formalin solution for histopathological examination. Then, samples were rinsed in running water, dehydrated in grades of ethanol and xylene, and embedded in paraffin. Paraffin blocks were cut into 4 µm slices, stained with hematoxylin–eosin and examined using the light microscope (Olympus CX21, Tokyo, Japan).

### 2.5. Bacteriological Tests

Tissue samples from the brain, lung, heart, liver, spleen, pancreas, kidney, mesenteric lymph node, and intestine, as well as nasal swabs, were collected post-mortem and tested using standard diagnostic methods. Columbia Agar with 5% sheep blood, MacConkey agar, Schaedler Agar with 5% sheep blood and Schaedler broth (GrasoBiotech, Owidz, Poland) were inoculated and incubated under aerobic, microaerophilic, and anaerobic conditions at 35.5 °C.

Identification of the isolates was performed by 16S rDNA sequencing (Genomed, Warsaw, Poland).

Antimicrobial susceptibility was tested using a disk diffusion method in accordance with the CLSI guidelines (CLSI, 2023).

### 2.6. Virological and Serological Tests

Samples for virological examination were collected post-mortem.

Initial screening of the nasal swab was performed using the Fluorecare^®^ (Milapharm, Ruislip, UK) test for the orthomyxovirus type A antigen.

Next, molecular diagnostics for SARS-CoV-2, IAV subtypes H1N1, H3N2, H5N1, and influenza virus type B (IBV) were performed. RNA was extracted from approximately 50 mg of each: lung, brain, spleen, liver, intestine, pancreas, heart, kidney, and mesenteric lymph node samples using a Total RNA Mini Kit (A&A Biotechnology, Grańsk, Poland), according to the manufacturer’s instructions. One-step reverse transcription real-time PCR (RT-qPCR) was performed using the CFx96 system (BioRad, Hercules, California, USA). Tests for SARS-CoV-2 were conducted with a commercially available Novel Coronavirus (COVID-19) Real Time Multiplex RT-PCR Kit (LifeRiver, San Diego, California USA), and with an in-house method described by Stefańska et al. for IAVs [25]. RT-qPCR results with a quantification cycle (*Cq*) of ≤ 35.00 were considered positive.
In order to detect antibodies to the feline immunodeficiency virus (FIV) and feline leukemia virus (FeLV) antigen serum samples were tested using the FIV Ab/FeLV Ag Combo Test (VetExpert, Łomianki, Poland).

## 3. Results

### 3.1. Clinical Course of the Disease and Treatment

Upon clinical examination, a fever of 40.5 °C, tachycardia, and mild dehydration were noted. Mucous membranes were pink and superficial lymph nodes were normal in size. No pathological crackles or murmurs were detected on auscultation. A tentative diagnosis of an upper respiratory tract infection was made and a combination of enrofloxacin (Enroxil, 5 mg/kg b.w. s.c., KRKA-POLSKA Sp. z o.o., Warsaw, Poland) and tolfenamic acid (Tolfine, 4 mg/kg b.w. s.c., Vetoquinol, Gorzów Wielkopolski, Poland) was administered. On the third day of treatment (19 June 2023), the cat’s condition deteriorated, and the patient was admitted to a 24 h veterinary clinic. The symptoms included fast and laborious breathing (100 breaths per minute), hypothermia (rectal body temperature of 34.6 °C), and tremors. The cat was placed in an oxygen cage with constant external heating. The treatment with enrofloxacin was maintained and supplemented with ceftriaxone (Biotraxon, 30 mg/kg b.w. i.v., Polpharma, Starogard Gdański, Poland), meloxicam (Metacam, 0.2 mg/kg b.w. s.c., Boehringer Ingelheim, Warsaw, Poland), and intravenous fluid therapy consisting of Ringer’s solution 50 mL/kg b.w./day with potassium chloride (Kalium Chloratum WZF 15%, Polfa Warsaw, Poland) 0.5 mEq/kg b.w./h, and a hepatoprotective drug containing ornithine (Ornipural, 2 mL/kg b.w., Vetoquinol, Gorzów Wielkopolski, Poland).

On the next day, generalized tremors, anisocoria (left pupil more dilated), and hypersensitivity to auditory, visual, and tactile stimuli developed. The cat gradually became semiconscious and began to drool. Subsequently, it experienced epileptic seizures, followed by respiratory and cardiac arrest and death (early morning of 21 June 2023).

Meanwhile, the owner reported that two other cats from the same cattery also died during this time, displaying severe dyspnea.

### 3.2. Blood Analysis

Blood tests revealed a considerable increase of activity of aspartate aminotransferase (AST; 2928 U/L; reference interval [RI]: ≤55 U/L), alanine aminotransferase (ALT; 2790 U/L; RI: ≤90 U/L), creatine kinase (CK; 8052 U/L; RI: ≤400 U/L), and lactate dehydrogenase (LDH; 2818 U/L; RI: 161–1051 U/L), increased total bilirubin concentration (6.4 mg/dL; RI: ≤0.6 mg/dL), and hypokalemia (2.8 mmol/L; RI: 4.1–5.6 mmol/L) accompanied by only mild hypernatremia (160 mmol/l; RI: 144–157 mmol/L) and hyperchloremia (127 mmol/L; RI: 102–118 mmol/L).

Hematological analysis was unremarkable except for a mild shift towards neutrophils in the differential leukogram: white blood cell count 14 G/L (RI: 5.5–19 G/L), red blood cell count 10.9 T/L (RI: 5–10 T/L), hemoglobin concentration 13.7 g/dL (RI: 9–15 g/dL), hematocrit 43% (RI: 28%–45%), platelet count 185 G/L (RI: 180–550 G/L), neutrophils 81% (RI: 50%–75%), lymphocytes 12% (RI: 20%–50%), monocytes 1% (RI: ≤4%), eosinophils 1% (RI: ≤6%), and basophils 5% (RI: ≤4%).

### 3.3. Ultrasound Examination and X-ray

The chest radiographs showed bilateral parenchymal densities in the anterior and middle lung fields (Figure 1 and Figure 2). Ultrasound examination revealed numerous subpleural consolidations in the anterior and middle lobes on the left lung and in the anterior lobe of the right lung. The size of consolidations varied from small lesions of 0.5–3 mm to lobular shred areas of approximately 10 × 13 mm (Figure 3). Additionally, numerous B lines were noted, indicating a “wet lung” image. Focused echocardiography revealed a trace amount of pericardial fluid, and a normal size of the heart chambers.

### 3.4. Gross and Microscopic Pathological Examination

Necropsy revealed hyperemia of many internal organs.

Microscopically, the most obvious lesions were present in the liver, lungs, intestine wall, and brain.

The liver displayed presence of numerous foci of necrosis, which were distributed in the whole organ (Figure 4A), while necrotic hepatocytes, cellular debris and proteinaceous material were localized centrally. In addition extravasated erythrocytes with peripherally localized inflammatory infiltrates consisting of lymphocytes and histiocytes were present (Figure 4B). Periportal inflammatory infiltrates with a mild intensity consisting of mononuclear cells were also observed.

In the lungs, atelectasis, hyperemia, interstitial and interalveolar hemorrhages with multifocal edema, and proteinaceous material in some alveoli were seen (Figure 5A). Furthermore multifocal accumulation of macrophages in the lumen of the alveoli (Figure 5B) and bronchioles was visible.

Necrotic and inflammatory lesions in myenteric plexuses of the small intestine were observed, characterized by vacuolization and necrosis of the ganglion cells as well as infiltration with lymphocytes and histocytes (Figure 6A,B).

The brain displayed perivascular infiltrates of lymphocytes with few histiocytes in the white and grey matter (Figure 7A) and the base of the choroid plexus (Figure 7B). Only a few foci of neuronal necrosis were observed thorough the brain.

Microscopic examination of samples of the remaining internal organs was unremarkable and did not reveal the presence of bacteria.

### 3.5. Bacteriological Examination

Pure cultures of abundant bacteria were obtained from the liver, kidney, lymph node, intestine, heart, and lung samples, as well as from the nasal swabs. Single colonies were cultured from the spleen and brain samples. No bacteria were found in the pancreatic tissue. Isolates from all samples were identified as *Enterococcus faecium* by 16S rDNA sequencing. The isolates were resistant to all antimicrobials tested: penicillin, amoxicillin-clavulanate, erythromycin, tetracycline, florfenicol, and enrofloxacin.

### 3.6. Virological and Serological Tests

The Fluorecare^®^–Milapharm test revealed orthomyxovirus type A antigen in the nasal swab.

Using RT-qPCR, samples from the lungs, liver, spleen, intestine wall, and mesenteric lymph node were found positive for A/H5N1 virus RNA, whereas samples from the brain, pancreas, heart, and kidney were negative. Exact results of the influenza A/H5N1 virus RT-qPCR are shown in Table 1. All samples were RT-qPCR negative for all other tested viruses, such as A/H1N1, A/H3N2, IBV, and SARS-CoV-2. Neither the FeLV antigen nor antibody against FIV were found in the serum.

## 4. Discussion

This case report provides the clinical description, along with pathological, virological, and bacteriological results of feline influenza A/H5N1 case, one of the series observed this summer in Poland. Meanwhile, in some other cats from this cluster of cases, A/H5N1 avian influenza was also confirmed, and the virus was characterized as belonging to the clade 2.3.4.4.b with mutations indicating a mammalian spread [31,32]. In our report, we presented, for the first time, detailed pulmonary ultrasound findings in a feline A/H5N1 virus infection as, in the past, ultrasonography was hardly used for lung examination. The cat’s medical history (including raw chicken meat consumption and outdoor access) and clinical presentation were typical of the disease caused by A/H5N1 virus in domestic cats. Thus far, few reports have consistently described this acute disease, characterized by fever, apathy and anorexia, rapidly progressing breathing difficulty, seizures, ataxia, and other neurological signs [21,26]. Additionally, following experimental intratracheal infection or ingestion of infected chicken, cats have exhibited fever from day 1 post-infection onwards, and signs such as apathy, protrusion of the nictitating membrane, conjunctivitis, and labored breathing by day 2 post-infection [19,21,33].

In the blood, significantly elevated activities of AST, ALT, CK, and LDH indicated considerable hepatocyte damage and potential muscle injury. Such increases suggest cytopathic effects induced by the virus, consistent with what one would expect from a A/H5N1 virus infection, as it is known that this agent has a broad cellular tropism. The increased total bilirubin level further confirms liver dysfunction or injury. Elevated red blood cell count combined with a hematocrit approaching the upper limit of the reference range may point to hemoconcentration, which can be attributed to dehydration or other secondary causes. Neutrophilia, with a corresponding decrease in the percentage of lymphocytes, indicates a systemic inflammatory response, commonly seen in viral infections. While these alterations are not unique to the A/H5N1 virus infection, they provide profound insight into the severity and systemic nature of this disease in cats.

The post-mortem examination of the cat revealed non-specific lesions indicative of acute circulatory insufficiency. Histopathological examination showed lesions in several internal organs, predominantly in the lungs, brain, liver, and intestinal wall. While these findings offer valuable insights into the extent of tissue involvement, it is important to emphasize that reaching a definitive diagnosis based on histopathology might not be possible. In the microbiological examination, *E. faecium* was cultured, while in the histopathological examination using hematoxylin and eosin staining, bacteria were not visualized. This discrepancy could be attributed to the limitations of the staining method, as well as to the distribution of bacteria that might not have been captured within the analyzed tissue sections. Pulmonary and brain lesions mirrored those observed in cats intratracheally inoculated with the IAV A/Vietnam/1194/2004 (A/H5N1), and the lesions in the intestinal plexuses resembled those observed after experimental alimentary infection. The initial testing for the orthomyxovirus type A antigen in the nasal swab proved positive, and post-mortem molecular investigation detected the RNA of the A/H5N1 virus in multiple organs but not in the brain. A possible explanation for the pathological changes in the brain, concurrent with the absence of virus detection, may indicate that the viral load in this tissue was too low to be detected, yet still sufficient enough to induce pathological alterations.

The detection of *E. faecium* in almost all tested organs suggests secondary septicemia, a common complication of generalized viral infections. *E. faecium* is a bacterium that naturally colonizes the gastrointestinal tract of cats. Recently, an increased resistance to antibiotics, such as ampicillin and vancomycin, has been observed in some stains [34,35]. While systemic infections in cats caused by this bacterium are relatively uncommon, when such an infection does occur, especially if it involves the central nervous system, the prognosis is typically unfavorable [36,37]. The multidrug resistance of the isolated strain would account for the lack of response to antimicrobial treatment in our patient. On the other hand, it is impossible to determine to what extent the A/H5N1 virus and/or the *E. faecium* infections were responsible for the emergence of the pathological symptoms in this case.

The source of the infection for the investigated cat remains unknown. The most common routes for cats to contract avian influenza virus appear to be direct contact with infected birds, consumption of raw poultry products, or hunting wild birds. These factors naturally put outdoor cats at higher risk. Our patient had outdoor access, and the A/H5N1 virus has been circulating in the wild bird population in Poland for quite some time. However, the last instances of A/H5N1 influenza in wild birds in the central Masovian Voivodeship, where the cat resided, were documented on 22 May 2023 in four dead black-headed gulls (Chief Vet. Off., https://www.wetgiw.gov.pl/nadzor-weterynaryjny/hpai, accessed on 14 July 2023). Given that the onset of the disease in the cat occurred almost one month later, and considering that the incubation period for experimental feline infections is approximately 1–2 days, this route of infection seems unlikely. However, it cannot be entirely ruled out, as on 12 June, four days before the first visit of our patient, A/H5N1 infection was confirmed in wild birds in two distant places, both about 200 km from Warsaw (around Reszel and Konin, according to the Chief Veterinary Office, https://www.wetgiw.gov.pl/nadzor-weterynaryjny/hpai, accessed on 27 August 2023). Even if such a distance seems to be a barrier for sick birds, subclinically infected ones could potentially carry the infection to distant areas undetected. Another potential, even though unconfirmed source of infection, could be the diet containing raw poultry products. The risk of such an event has been assessed in the past. A study by Golden et al. [38] estimated that in a 20,000-bird house with a single infected bird, the probability that an A/H5N1 HPAIV-infected flock would be detected before slaughter is approximately 94%, as the virus spreads rapidly and the mortality is quickly increased. The only infected flock likely to reach slaughter undetected would be one that was infected roughly 3.5 days prior to shipment. These estimates suggest that the probability of an infected, undetected flock going to slaughter is low, although not zero [38]. Similarly, another study estimated for Germany the risk of an undetected infected flock being slaughtered at virtually zero. However, when more pessimistic ranges of expert opinions are taken into account, up to 23 such instances per year might occur [39]. Indeed, in Germany, such an event probably occurred at least once. The HPAI A/H5N1 virus caused mortality in backyard chickens in three separate holdings spaced 80–120 km apart, although there were no reported cases of infection in wild birds or in poultry in that area for the preceding four months. However, all these chickens had access to uncooked offal from commercial deep-frozen duck carcasses purchased by the owners at a supermarket [40].

The cat’s owner reported that almost concurrently with our patient, two other cats in his household developed severe respiratory symptoms and died. By analogy, suspicion of A/H5N1 virus infection can be made in them, However, it should obviously be treated with caution. Experimental infections with A/H5N1 virus have demonstrated that both respiratory and alimentary infections can lead to the transmission of the virus to other cats [19,33]. However, asymptomatic cats shed only minimal amounts of the virus [41]. Moreover, in an experiment investigating interspecies transmission of the A/H5N1 virus infection, cats did not become infected when cohabiting with or sharing bowls with infected dogs and vice versa [33]. In the household of our patient, all three cats had outdoor access and used to eat raw chicken meat, so it is likely that if the A/H5N1 virus was indeed the cause of death of the two other cats, they became infected from the same external source rather than from each other.

On very rare occasions, HPAIVs can infect humans and induce a severe disease with high mortality. Since the emergence of the A/H5N1 virus in 1996, approximately 870 human cases resulting from contact with birds have been registered worldwide (https://cdn.who.int/media/docs/default-source/influenza/human–animal-interface-risk-assessments/2022_nov_tableh5n1.pdf?sfvrsn=babfcad1_1&download=true, accessed on 27 August 2023). However, feline A/H5N1 infections raise concerns that adaptation of avian viruses to cats—animals living in close contact with humans, and hunting birds—could represent the first step towards potential human-to-human transmission of HPAIVs, potentially resulting in a new pandemic. The clade 2.3.4.4b is linked to the Gs/Gd H5N1 influenza virus as a progenitor, with genetic mutations and adaptations over time leading to the emergence of the latter. This evolutionary process involved changes in the viral genome which contribute to alterations in various aspects of the virus, including its virulence, host specificity, and potential for replication among different species [42]. Infections of mammals with the 2.3.4.4b clade are more frequent than with other HPAIV clades. Nearly all mammalian strains belonging to clade 2.3.4.4b were isolated within last two to three years, even though HPAI outbreaks have been occurring worldwide for over 20 years. According to Rabalski et al., the most plausible explanation for this situation is that the clade 2.3.4.4b, with point mutations, is indicative of adaptations to mammalian hosts [31]. Furthermore cats have both alpha-2,6 and alpha-2,3 receptors in their trachea and lungs, and thus they could potentially serve as intermediate hosts for influenza viruses, both of avian and human origin [22,43]. While it is impossible to control HPAI in wild birds, which serve as a mobile reservoir for IAV genes, cats on endemic areas should be prevented from going outdoors, hunting, and feeding on raw meat.

As for now, there have been no documented cases of human infections with the A/H5N1 virus stemming from contact with infected cats, despite the proximity between these animals and their caregivers. Current assessments indicate a low risk of human infection in the general population, and a low-to-moderate risk among individuals in close contact with cats. The World Health Organization has emphasized that this risk could evolve due to viral mutations (https://www.who.int/emergencies/disease-outbreak-news/item/2023-DON476, accessed on 27 August 2023; https://www.efsa.europa.eu/en/news/avian-influenza-efsa-recommends-increased-surveillance, accessed on 27 August 2023). Notably, the risk is constantly evaluated and managed within a rigorous legal framework. Specifically, governmental bodies, in the context of mitigating risks associated with HPAI, operate under guidelines delineated in Commission Delegated Regulation (EU) 2020/687, dated 17 December 2019. This serves as a supplementary document to the Regulation (EU) 2016/429 of the European Parliament and of the Council, laying out specific provisions for the prevention and containment of select diseases. In Poland, these guidelines are further specified through a ministerial directive, issued on 23 March 2023, which introduces the ‘National Program for the Detection of Avian Influenza Virus Infections in Poultry and Wild Birds’ for the calendar year 2023 (Journal of Laws 2023, item 618). This cohesive regulatory structure underscores the commitment of governmental agencies to safeguard both public and animal health through evidence-based policy measures.

## Figures and Tables

**Figure 1 microorganisms-11-02263-f001:**
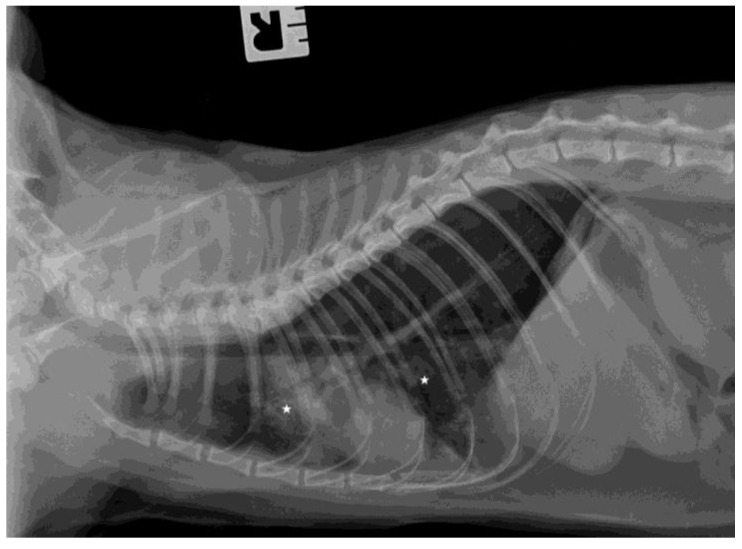
The thoracic X-ray image in the lateral right-left (RL) position. Parenchymal densities are visible in the anterior and middle fields of the lungs (asterisks indicate areas of lesions in the lungs).

**Figure 2 microorganisms-11-02263-f002:**
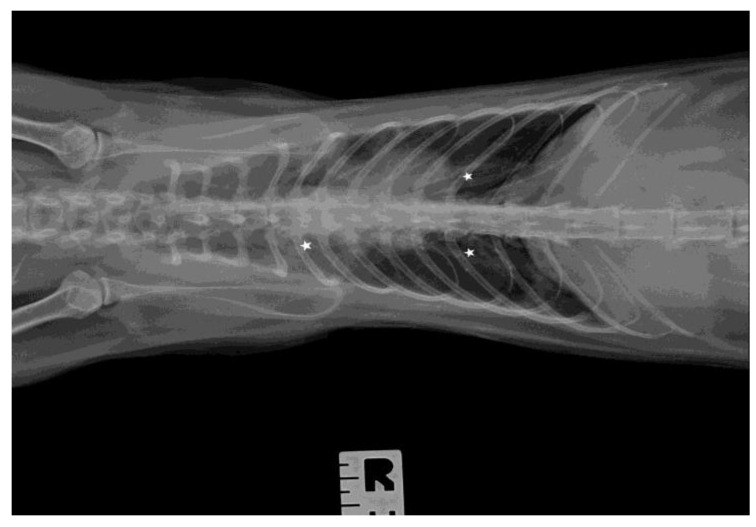
The thoracic X-ray image in the anteroposterior (AP) position. Parenchymal densities are visible in the middle and caudal fields of the lungs (asterisks indicate areas of lesions in the lungs).

**Figure 3 microorganisms-11-02263-f003:**
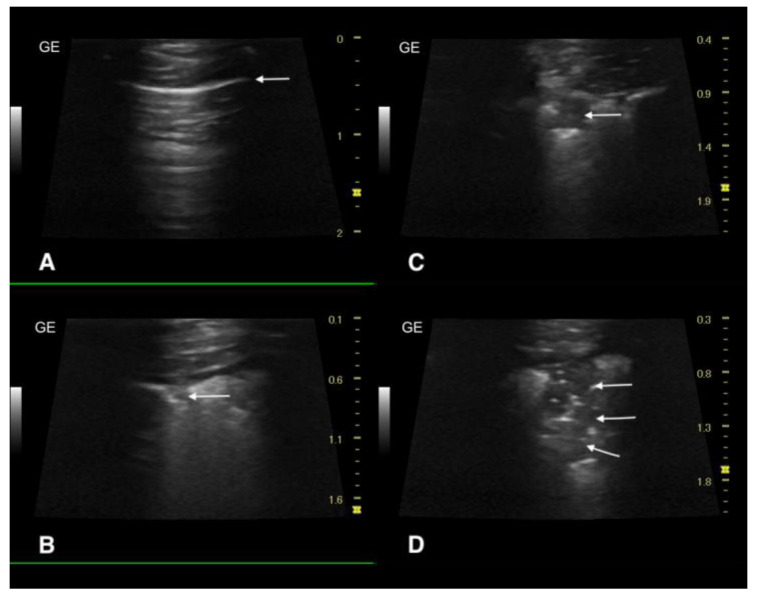
Ultrasonographic examination of the lungs. (**A**) Image of a healthy lung filled with air, the white arrow indicates the normal, smooth pleural line, (**B**) A small subpleural consolidation, with a diameter of approximately 1.5 mm (white arrow), (**C**) A subpleural consolidation of approximately 0.5 cm in size (white arrow), (**D**) Image of a consolidation involving a portion of the lung to the depth of approximately 1.8 cm, an aerated area of the affected lung is visible (white arrows).

**Figure 4 microorganisms-11-02263-f004:**
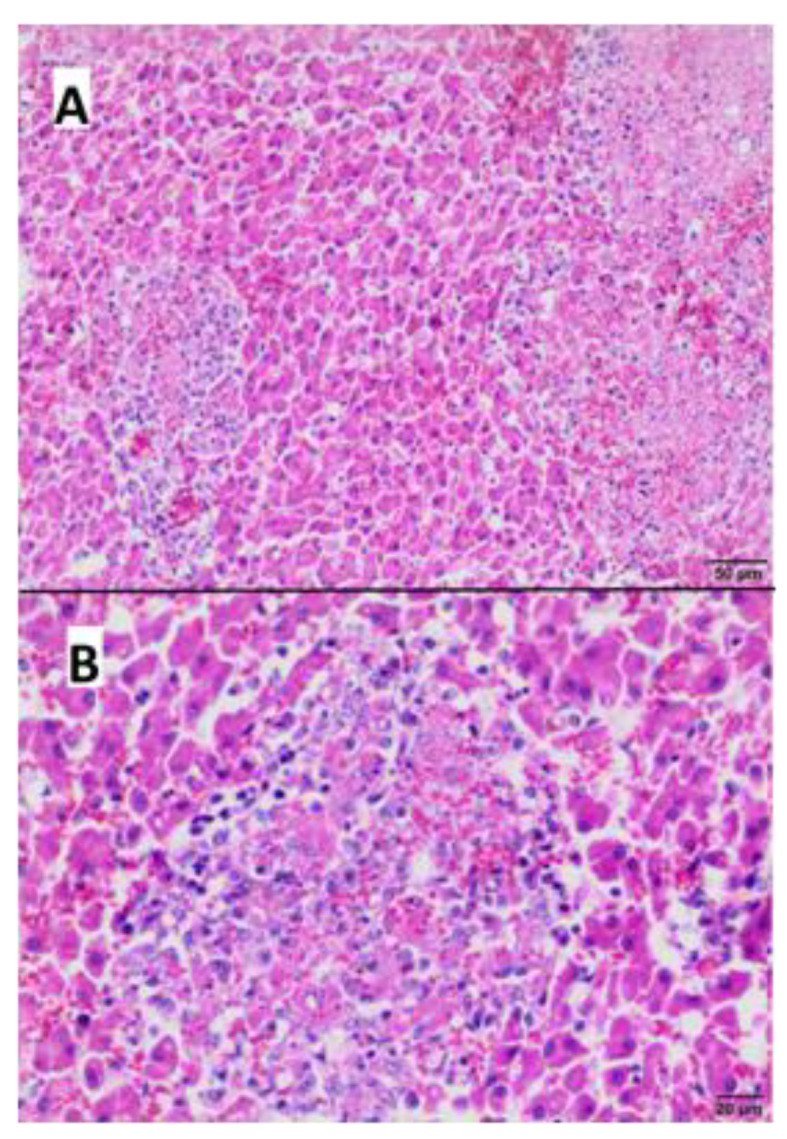
Histopathological examination of the liver—(**A**,**B**) areas of necrosis with inflammatory infiltrate composed of lymphocytes and histiocytes (Hematoxylin-eosin staining; (**A**) bar 50 µm; (**B**) bar 20 µm).

**Figure 5 microorganisms-11-02263-f005:**
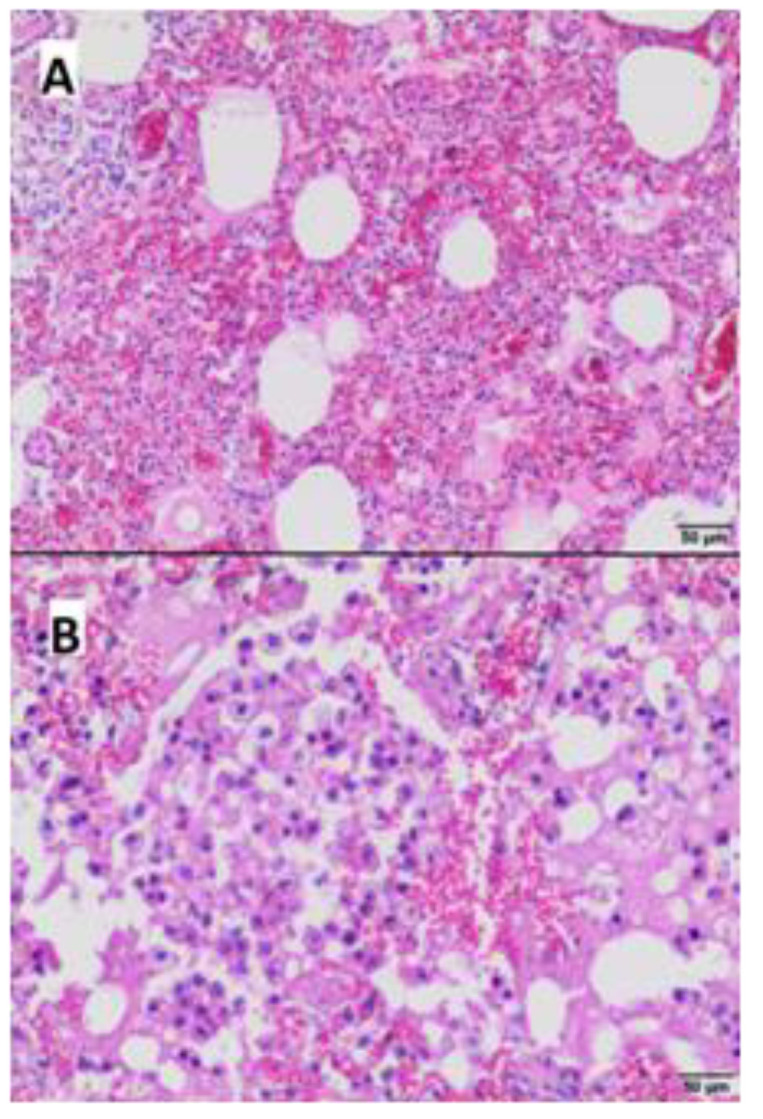
Histopathological examination of the lungs. (**A**) atelectasis, hyperemia, interstitial and alveolar hemorrhages. (**B**) macrophages within the alveolar lumen (Hematoxylin-eosin staining; (**A**) bar 50 µm; (**B**) bar 20 µm).

**Figure 6 microorganisms-11-02263-f006:**
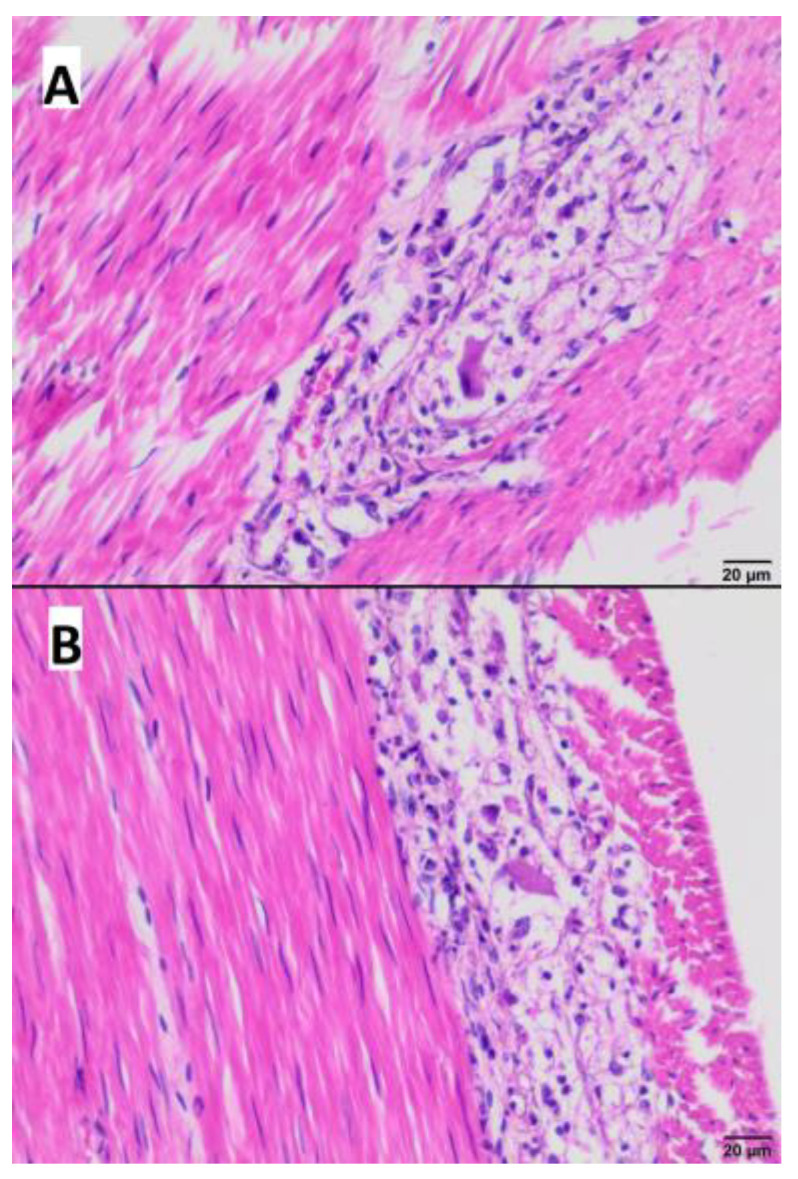
Histopathological examination of the intestinal wall. (**A**,**B**) vacuolization and necrosis of myenteric plexus cells with inflammatory infiltrate consisting of lymphocytes and histiocytes (Hematoxylin-eosin staining; (**A**) bar 20 µm; (**B**) bar 20 µm).

**Figure 7 microorganisms-11-02263-f007:**
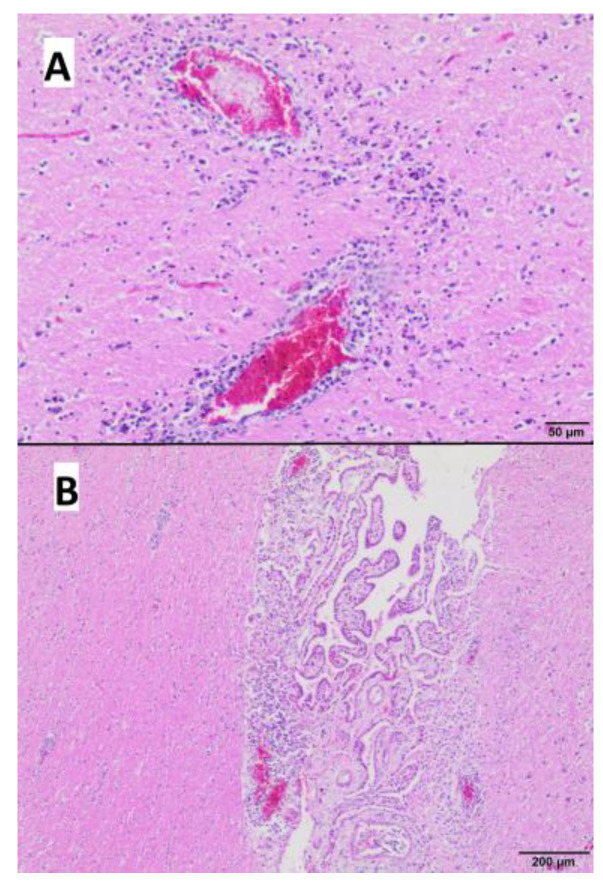
Histopathological examination of the brain. (**A**) perivascular infiltrates of lymphocytes and histiocytes in the white and gray matter. (**B**) perivascular infiltrate of lymphocytes and few histiocytes at the base of the choroid plexus (Hematoxylin-eosin staining; (**A**) bar 50 µm; (**B**) bar 200 µm).

**Table 1 microorganisms-11-02263-t001:** Results of RT-qPCR for influenza A/H5N1 virus genes.

Clinical Sample	Result	*Cq* for H5 Gene	*Cq* for N1 Gene
spleen	(+)	31.58	31.33
intestine	(+)	32.14	31.84
liver	(+)	33.21	33.07
lung	(+)	34.02	33.56
mesenteric lymph node	(+)	34.71	33.92
pancreas	(−)	-	-
heart	(−)	-	-
kidney	(−)	-	-
brain	(−)	-	-

## Data Availability

No new data were created or analyzed in this study. Data sharing is not applicable to this article.

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
