# Peer review of "A Fatal A/H5N1 Avian Influenza Virus Infection in a Cat in Poland"

_microorganisms, 2023, doi:10.3390/microorganisms11092263_

Round 1
Reviewer 1 Report
Szalus-Jordanow et al. has reported a case of domestic cat in Poland that has succumbed to HPAIV 2.3.4.4b infection. The clinical and pathological findings are useful and necessary to be disseminated in the wider community to raise awareness among veterinarians and pathologists of the disease presentation. However, there are significant gaps in the manuscript that will need to be addressed properly for further consideration of the paper.
1. Lack of molecular confirmation of a highly pathogenic H5 (a generic H5 is not acceptable) that the virus is consistent with the existing circulating H5Nx in Europe.
2. Anatomical and clinical descriptions as well as interpretations are poor.
3. Viral immunohistochemistry not used to corroborate histopathological findings (there are numerous paper for 2.3.4.4.b that have both H&E and IHC). This could have been useful if a specific HP H5 cannot be confirmed/achieved, as neurotropism or vascular tropism is pretty much consistent with a HPAI presentation.
4. Grammar and styles will require significant review and overhaul, as the write up often gave wrong/inappropriate impression of the work/interpretation, and can be easily taken out of context as a result.
Below are specific feedbacks:
Abstract
Please use the appropriate pathological terminologies to describe vascular lesions of the brain. And as discussed below, morphological diagnoses should be provided for other organs within the abstract.
‘On very rare occasions the A/H5N1 strain can cross the interspecies barrier, leading to disease and substantial mortality in cats. This is especially true when these animals are exposed to infected birds or contaminated raw meat. Nevertheless, the definitive source of infection could not be identified in this case.’
This sentence needs to be more succinct. The key takeaway is that the definitive sources are either exposure to infected birds or contaminated raw meat.
‘More similar cases have been observed at the same time in different regions of Poland.’
This sentence is not appropriate within the abstract as it is not data generated from your paper per se. Please remove.
Introduction
Line 53 – ‘each HPAI virus (HPAIV) eventually became extinct’. Extinct probably not the best choice of word, they reassort and mutate to leading to a subclade of virus.
Line 69 – ‘The epidemic continued into 2023.’ The virus re-emerged in Europe rather than continued from previous season.
Line 70 – The authors should establish the context that the initial wave of HPAIV 2.3.4.4b in Europe, with confirmed mammalian cases, was observed in canids and seals (refer to Floyd et al. 2021).
Furthermore, the absence of any reference to the presumed origin of the 2.3.4.4b strain, namely the Gs/Gd H5N1, is concerning and indicates a lack of comprehension of the virological intricacies associated with the present 2.3.4.4b virus.
2.1 Clinical findings.
The provided information should be categorized under the "Results" section rather than within the "Materials and Methods" section. What is useful within M&M is the span of dates where data was collected.
Additionally, details about the cat's history of vaccination need to be included. Clarification is needed regarding whether the cat was under the care of a private owner at the time of reporting or was located in a cattery (as a business entity), especially considering the mention of other cats that died in a cattery. The description of the epidemiological context should be explicit and clear.
2.2
It is unclear what you mean by dry tube. Do you mean a serum clot tube?
2.4
Please mention the type of stain preparation of a histopathological evaluation.
2.6 Virological test
The text does not include any mention of testing for common endemic cat viruses, such as Feline herpesvirus, FIV, and FeLV. This absence could be an unintended omission from the text, but it may also be perceived as a potential oversight in the disease investigation process.
3.1 and 3.3 Blood analysis and pathological examination
The description of clinical parameters (hematology and biochemistry) as well as anatomical pathology should be enhanced to incorporate appropriate terminologies and interpretations. For instance, "decreased potassium concentration" should be revised to "hypokalemia."
It is important to specify the preservation method of the carcass (frozen or fresh). It should be noted that frozen carcasses may exhibit a red appearance due to hemoglobin imbibition from lysed thawed red blood cells. Clarification on this point is needed. Furthermore, the absence of other gross findings is notable, and some changes should be documented based on the histological description.
The histological description necessitates significant corrections and standardization according to the conventions of histopathological reporting. For instance, the phrase "Lungs: atelectasis, hyperemia, interstitial and intraalveolar hemorrhages, with multifocal edema, and proteinaceous material in some alveoli (Figure 5A), multifocal accumulation of macrophages in the lumen of alveoli (Figure 5B) and bronchioles" should be revised to align with accepted reporting standards. For example "The lungs displayed atelectasis, hyperemia, interstitial and intraalveolar hemorrhages, along with multifocal edema and the presence of proteinaceous material in some alveoli (Figure 5A). Furthermore, multifocal accumulation of macrophages within the lumen of alveoli (Figure 5B) and bronchioles was observed."
In cases of inflammation, it is crucial to specify the exact cell type involved, such as lymphoplasmacytic or histiocytic. The term "mononuclear inflammation" by itself is not sufficient. Additionally, it is recommended to provide a morphological diagnosis, for instance, "moderate, multifocal, interstitial pneumonia."
Given the isolation of bacteria from multiple organs, the histological description should also mention if bacterial structures were detected through microscopy and whether special stains were employed during evaluation. If bacteria were not detected through microscopy, an explanation for this outcome should be provided.
The histological images require proper annotations.
3.2 - Radiological and ultrasonographical description is great. Please add some arrows / annotations to the images and update within caption.
3.4 Bacteriology
‘Numerous bacteria, in pure culture, were isolated from the liver, kidney, lymph node, intestine, heart, and lung samples, as well as from the nasal swabs.’
Do you mean abundant instead of numerous? My first impression is that you have yielded multiple species of bacteria.
3.5 Virology
Please provide Cq values for the PCR+ tissues and swabs.
4. Discussion
‘However, feline A/H5N1 infections raise concerns that adaptation of avian viruses to cats – animals living in close contact with humans and hunting birds – could represent the first step toward potential human-to-human transmission of HPAIVs, potentially resulting in a new pandemic.’
To date, there is limited evidence of feline adaptation to influenza. Early molecular studies have suggested potential susceptibility and transmission of influenza in species like mink, ferrets, and foxes. For a comprehensive understanding of this topic, it is recommended to refer to relevant sources from organizations such as the European Food Safety Authority (EFSA) and the Centers for Disease Control and Prevention (CDC)/Animal and Plant Health Inspection Service (APHIS) data. Proper citation of these sources should be included to support the discussion on feline adaptation to influenza.
Grammar and styles will require significant review and overhaul, as the write up often gave wrong/inappropriate impression of the work/interpretation, and can be easily taken out of context as a result.
Author Response
Comments and Suggestions for Authors
Szalus-Jordanow et al. has reported a case of domestic cat in Poland that has succumbed to HPAIV 2.3.4.4b infection. The clinical and pathological findings are useful and necessary to be disseminated in the wider community to raise awareness among veterinarians and pathologists of the disease presentation. However, there are significant gaps in the manuscript that will need to be addressed properly for further consideration of the paper.
Reviewer: Lack of molecular confirmation of a highly pathogenic H5 (a generic H5 is not acceptable) that the virus is consistent with the existing circulating H5Nx in Europe.
Our answer: Thank you for your thoughtful evaluation of our manuscript and for highlighting the importance of providing specific molecular confirmation of the highly pathogenic H5. We acknowledge the significance of molecular data in supporting our findings and understand your concern. However, we believe that, in the interest of healthcare professionals and cat caretakers, it is important to publish this influenza case in a cat in Poland to make them aware of this highly deadly disease. Our primary objective with this publication was to document the disease progression, using ultrasonographic techniques to assess the severity of lung changes in animals with influenza, as is done in the human medicine, pathological examination, and testing results due to the rarity of such cases in Europe.
Furthermore, the multiplex RT-qPCR reaction used in our study described by Stefańska et al. allows for the simultaneous detection and differentiation of the N1 and N2 genes as well as the H1, H3 and H5 genes of influenza A viruses. These simultaneous three duplex real-time PCR assays make it possible to determine whether we are dealing with the presence of A/H5N1 RNA. We acknowledge and concur with your observation. Our PCR method was indeed not tailored to differentiate between HPAI and LPAI H5N1 strains. In light of this, we have removed the mentioned statement from our manuscript to ensure accuracy and avoid any potential misconceptions.
However, we'd like to provide some context which may be of interest. While our findings are not definitive, we find it plausible, based on external evidence, that the virus strain detected belong to the clade 2.3.4.4.b. This is supported by the findings of Prof. Krzysztof Pyrć's team, who confirmed this clade in two isolates from cats from Poland. Nonetheless, we agree that until we have concrete sequencing data, any such association remains speculative.
During the review of our study, apart from Professor Pyrc's team's work mention above (Robalski et al.), another paper concerning the sequencing of cases of A/H5N1 influenza virus in Poland was also published (our case was not included). All the examined cases belonged to the clade 2.3.4.4b of the influenza virus. We have added appropriate citations to the manuscript.
Reviewer: Anatomical and clinical descriptions as well as interpretations are poor.
Our answer: We have carefully considered the suggestions provided in all the reviews and have made the necessary revisions accordingly.
Reviewer: Viral immunohistochemistry not used to corroborate histopathological findings (there are numerous papers for 2.3.4.4.b that have both H&E and IHC). This could have been useful if a specific HP H5 cannot be confirmed/achieved, as neurotropism or vascular tropism is pretty much consistent with a HPAI presentation.
Our answer: Unfortunately, it was impossible to perform immunofluorescence assays (DIF or IIF) in tissue samples due to the lack of available appropriate tests for influenza viruses in our laboratory. The standard diagnostic procedure for suspected influenza virus infections includes rapid antigen tests as a screening method and confirmation with a molecular biology test. Molecular studies for the determination of variants of influenza viruses are mainly performed by other reference units. Such a test, according to the method described by Stefańska et al., was used in our research only because it was developed in cooperation with two researchers from our team.
Reviewer: Grammar and styles will require significant review and overhaul, as the write up often gave wrong/inappropriate impression of the work/interpretation, and can be easily taken out of context as a result.
Our answer: We have incorporated all the suggested revisions and have submitted the article to a professional for grammar and spelling checks to ensure its accuracy and quality.
Below are specific feedbacks:
Abstract
Reviewer: Please use the appropriate pathological terminologies to describe vascular lesions of the brain. And as discussed below, morphological diagnoses should be provided for other organs within the abstract.
Our answer: done
Reviewer: ‘On very rare occasions the A/H5N1 strain can cross the interspecies barrier, leading to disease and substantial mortality in cats. This is especially true when these animals are exposed to infected birds or contaminated raw meat. Nevertheless, the definitive source of infection could not be identified in this case.’
This sentence needs to be more succinct. The key takeaway is that the definitive sources are either exposure to infected birds or contaminated raw meat.
Our answer: done
Reviewer: ‘More similar cases have been observed at the same time in different regions of Poland.’
This sentence is not appropriate within the abstract as it is not data generated from your paper per se. Please remove.
Our answer: done
Introduction
Reviewer: Line 53 – ‘each HPAI virus (HPAIV) eventually became extinct’. Extinct probably not the best choice of word, they reassort and mutate to leading to a subclade of virus.
Our answer: corrected
Reviewer: Line 69 – ‘The epidemic continued into 2023.’ The virus re-emerged in Europe rather than continued from previous season.
Our answer: corrected
Reviewer: Line 70 – The authors should establish the context that the initial wave of HPAIV 2.3.4.4b in Europe, with confirmed mammalian cases, was observed in canids and seals (refer to Floyd et al. 2021).
Our answer: done
Reviewer: Furthermore, the absence of any reference to the presumed origin of the 2.3.4.4b strain, namely the Gs/Gd H5N1, is concerning and indicates a lack of comprehension of the virological intricacies associated with the present 2.3.4.4b virus.
Our answer: added
2.1 Clinical findings.
Reviewer: The provided information should be categorized under the "Results" section rather than within the "Materials and Methods" section. What is useful within M&M is the span of dates where data was collected.
Our answer: done
Reviewer: Additionally, details about the cat's history of vaccination need to be included. Clarification is needed regarding whether the cat was under the care of a private owner at the time of reporting or was located in a cattery (as a business entity), especially considering the mention of other cats that died in a cattery. The description of the epidemiological context should be explicit and clear.
Our answer: done
2.2
Reviewer: It is unclear what you mean by dry tube. Do you mean a serum clot tube?
Our answer: yes, corrected
2.4
Reviewer: Please mention the type of stain preparation of a histopathological evaluation.
Our answer: done
2.6 Virological test
Reviewer: The text does not include any mention of testing for common endemic cat viruses, such as Feline herpesvirus, FIV, and FeLV. This absence could be an unintended omission from the text, but it may also be perceived as a potential oversight in the disease investigation process.
Our answer: done
3.1 and 3.3 Blood analysis and pathological examination
Reviewer: The description of clinical parameters (hematology and biochemistry) as well as anatomical pathology should be enhanced to incorporate appropriate terminologies and interpretations. For instance, "decreased potassium concentration" should be revised to "hypokalemia."
Our answer: done
Reviewer: It is important to specify the preservation method of the carcass (frozen or fresh). It should be noted that frozen carcasses may exhibit a red appearance due to hemoglobin imbibition from lysed thawed red blood cells. Clarification on this point is needed. Furthermore, the absence of other gross findings is notable, and some changes should be documented based on the histological description.
Our answer: added
Reviewer: The histological description necessitates significant corrections and standardization according to the conventions of histopathological reporting. For instance, the phrase "Lungs: atelectasis, hyperemia, interstitial and intraalveolar hemorrhages, with multifocal edema, and proteinaceous material in some alveoli (Figure 5A), multifocal accumulation of macrophages in the lumen of alveoli (Figure 5B) and bronchioles" should be revised to align with accepted reporting standards. For example "The lungs displayed atelectasis, hyperemia, interstitial and intraalveolar hemorrhages, along with multifocal edema and the presence of proteinaceous material in some alveoli (Figure 5A). Furthermore, multifocal accumulation of macrophages within the lumen of alveoli (Figure 5B) and bronchioles was observed."
Our answer: corrected
Reviewer: In cases of inflammation, it is crucial to specify the exact cell type involved, such as lymphoplasmacytic or histiocytic. The term "mononuclear inflammation" by itself is not sufficient. Additionally, it is recommended to provide a morphological diagnosis, for instance, "moderate, multifocal, interstitial pneumonia."
Our answer: corrected. While these findings offer valuable insights into the extent of tissue involvement, it is important to emphasize that reaching a definitive diagnosis based on histopathology might not be possible - we’ve added this information.
Reviewer: Given the isolation of bacteria from multiple organs, the histological description should also mention if bacterial structures were detected through microscopy and whether special stains were employed during evaluation. If bacteria were not detected through microscopy, an explanation for this outcome should be provided.
Our answer: done, the information has been supplemented in the Results section and some in the Discussion section.
Reviewer: The histological images require proper annotations.
Our answer: corrected.
Reviewer: 3.2 - Radiological and ultrasonographical description is great. Please add some arrows / annotations to the images and update within caption.
Our answer: done
3.4 Bacteriology
Reviewer: ‘Numerous bacteria, in pure culture, were isolated from the liver, kidney, lymph node, intestine, heart, and lung samples, as well as from the nasal swabs.’
Do you mean abundant instead of numerous? My first impression is that you have yielded multiple species of bacteria.
Our answer: yes, corrected.
3.5 Virology
Reviewer: Please provide Cq values for the PCR+ tissues and swabs.
Our answer: done
- Discussion
Reviewer: ‘However, feline A/H5N1 infections raise concerns that adaptation of avian viruses to cats – animals living in close contact with humans and hunting birds – could represent the first step toward potential human-to-human transmission of HPAIVs, potentially resulting in a new pandemic.’
To date, there is limited evidence of feline adaptation to influenza. Early molecular studies have suggested potential susceptibility and transmission of influenza in species like mink, ferrets, and foxes. For a comprehensive understanding of this topic, it is recommended to refer to relevant sources from organizations such as the European Food Safety Authority (EFSA) and the Centers for Disease Control and Prevention (CDC)/Animal and Plant Health Inspection Service (APHIS) data. Proper citation of these sources should be included to support the discussion on feline adaptation to influenza.
Our answer: done

Reviewer 2 Report
This study reports the first case of HPAI in a cat in Poland. The information of symptoms and the process of diagnosis in this report will be useful to take measures to HPAI in cat because the cat infections with HPAIV are rare. However, the novelty may be little poor because the details of symptoms including anatomicopathological results were likely to past reports. The addition of the epidemiologic data of this case will make this report worth.
Please add some clarifications and fix by following comments.
■line 43
You wrote that cats are resistant to influenza. Which does this mean so-called "feline influenza" has not been circulating inside cats’ community or cats are not susceptive to any influenza virus? Though Influenza virus rarely infects over species-barrier, GsGd H5 viruses sporadically have infected to some mammalian species, including human. 46 H5 viruses were isolated from Felidae, except isolated in Poland, based on virus information in GISAID database, whereas 19 in dog and 59 in pig. Therefore, Felidae does not seem to be unsusceptive to H5 virus more than the other mammalian species. I thought that Felidae was considered susceptive to HPAI at latest middle 2000s because of the cases in Thailand.
■line 43-57
This section is incoherent and makes confusing. You described the different virus in each sentence. For example, FIV in 1st, IFV in 2nd, FIV in 3rd and 4th, IFV in 5th, AIV in 6th...
You had better describe the only FIV and IFV in the first section, and AIV in the second section.
■line 55-56
GsGd H5 viruses were identified in "Guangdong" in 1996, weren't it?
■line 69
The situation of HPAI in 2022-2023 season is important more than that in 2021-2022 season because it occurred the case in this manuscript in June, 2023. You should describe it.
■line 70-80
Why do you mention to only H5N1 not H5Nx about infection to cats? What is the difference between H5N1 and H5Nx though the cases of the infection to cats with H5N6 have reported in China, 2014-16. As all researchers of AIV know, the origin of N1 on the early H5N1 virus differs completely from that on 2.3.4.4b H5N1.
In addition, are the cases in cats the relatively major case among “mammalian case with GsGd H5 viruses", aren't it, though the mammalian cases are rare?
And, mammalian infection with 2.3.4.4b HPAIV were more frequency than with other clade HPAIV, and the number of the registered strain(295) in GISAID is over half of total number(452) of HPAIV derived from mammals except human. The almost all mammalian strains in 2.3.4.4b were isolated in two to three years though HPAI outbreaks have occurred worldwide over 20 years. How do you think the relationship between this case in cat and these facts?
■line 244
You should discuss that there were lesions in the brain but RNA was not detected in that. Furthermore, the immunohistological results will make readers more understanding if there are.
■line 250-255
You should explain the pathogenicity of E. faecium and discuss its contribution to this dead case. As you said the responsibility of E. faecium for the emergence of the pathological lesions is impossible to determine, but you can discuss. Though E. faecium is known to the opportunistic pathogen in the field, it should be also explained in the manuscript.
■line 256-283
This is the high-profile section in this manuscript. But, we cannot understand which the risk of natural infection was high or low because the situation of HPAI in this season was not shown. You described that HPAIV in wild birds was lastly detected in Masovian Voivodeship on May 22, 2023. What about in the neighboring Voivodeship such as Lubelskie Voivodeship? The cat resided in the central or outlying in Masovian Voivodeship? Does cats usually catch wild waterfowl or small mammals such as mouses? In this version of manuscript I receive the impression that the cat has been infected with HPAIV from the fed made of infected poultries not "natural" such as from wild birds. The possibility of the infection from "natural" should be more considered because the evidences of that from the fed or others were absent.
Recently, the HPAI case in cat in Korea was reported to be caused by the fed made of infected poultries.(https://en.yna.co.kr/view/AEN20230802003000320) Thus, even now the HPAI infections by improper feds can be occurred. But, your explanation is unfair because the reference you cited is old. You should be described that EU or Poland government takes measures to reduce the risk of HPAI spread after the incident you cited. For example, shipped meats from the farm must be recalled back to 21 days if HPAI occurred in the farm by EU regulation.
Author Response
Reviewer: This study reports the first case of HPAI in a cat in Poland. The information of symptoms and the process of diagnosis in this report will be useful to take measures to HPAI in cat because the cat infections with HPAIV are rare. However, the novelty may be little poor because the details of symptoms including anatomicopathological results were likely to past reports. The addition of the epidemiologic data of this case will make this report worth.
Please add some clarifications and fix by following comments.
Our answer: We understand your perspective about the similarity in symptom details to past reports. However, such a cluster of cases is definitely unusual, thus we'd like to alert veterinarians also in other countries that such an outbreak is possible.
Historically, while individual cases of highly pathogenic avian influenza in cats have been acknowledged, the scenario Poland experienced in recent weeks is unparalleled on a global scale. Generally, cases of this infection in companion animals within a country tend to be isolated incidents, often associated with poultry farms and concurrent cases of influenza in poultry or local wild waterfowl.
Contrastingly, Poland witnessed numerous outbreaks in domestic cats within a short span, scattered across vast distances, involving both outgoing and home-bound felines. Regrettably, the full magnitude of cases remains unknown, as many cats succumbed before influenza could be confirmed. This situation swiftly escalated, fueled by internet reports, leading to panic among cat owners. The rapid progression of the disease, coupled with the absence of easy access to animal influenza diagnostics and a dearth of management guidelines for such diseases in companion animals, exacerbated the crisis. Furthermore, the veterinary community faced immense pressure distressed pet owners and the media. The situation grew grimmer as speculative voices emerged, suggesting the extermination of cats, particularly strays.
Given the gravity and novelty of this situation, we found it imperative to detail this clinical case exhaustively. Our objective is twofold: to facilitate the recognition of such cases in the future and to advocate for the establishment of guidelines or regulations for similar events. We genuinely hope that other nations will never have to navigate such a multifaceted crisis, but should it occur, a robust understanding of the clinical progression is paramount.
In conclusion, while we acknowledge the resemblance of some of our findings to previous reports, we firmly believe that the context and implications of our study elevate its relevance and potential impact. We trust you'll consider the broader implications of our work in the context of the unprecedented events in Poland.
■line 43
Reviewer: You wrote that cats are resistant to influenza. Which does this mean so-called "feline influenza" has not been circulating inside cats’ community or cats are not susceptive to any influenza virus? Though Influenza virus rarely infects over species-barrier, GsGd H5 viruses sporadically have infected to some mammalian species, including human. 46 H5 viruses were isolated from Felidae, except isolated in Poland, based on virus information in GISAID database, whereas 19 in dog and 59 in pig. Therefore, Felidae does not seem to be unsusceptive to H5 virus more than the other mammalian species. I thought that Felidae was considered susceptive to HPAI at latest middle 2000s because of the cases in Thailand.
Our answer: Upon revisiting the text, we recognize the potential ambiguity in our phrasing, specifically where we mentioned, "For a long time, cats were considered resistant to influenza." Our intent was to convey that historically, and during much earlier times, there was a prevailing belief about cats being resistant due to a lack of documented cases. We did not intend to suggest that cats are inherently resistant to influenza or that there haven't been recent findings contradicting this belief. As you rightly pointed out, there have indeed been cases that indicate Felidae's susceptibility to HPAI, especially in the mid-2000s and later.
Given the potential for confusion, and in the interest of maintaining clarity and accuracy in the text, we are inclined to either amend this statement or remove it entirely, in line with your suggestion. Your feedback is invaluable to us, and we strive to ensure the information we present is both accurate and easily comprehensible to readers.
■line 43-57
Reviewer: This section is incoherent and makes confusing. You described the different virus in each sentence. For example, FIV in 1st, IFV in 2nd, FIV in 3rd and 4th, IFV in 5th, AIV in 6th...
You had better describe the only FIV and IFV in the first section, and AIV in the second section.
Our answer: This section of the Introduction has been modified. Used abbreviations have been harmonized, with their full forms delineated within the manuscript.
■line 55-56
Reviewer: GsGd H5 viruses were identified in "Guangdong" in 1996, weren't it?
Our answer: This has been clarified.
■line 69
Reviewer: The situation of HPAI in 2022-2023 season is important more than that in 2021-2022 season because it occurred the case in this manuscript in June 2023. You should describe it.
Our answer: The information pertaining to the HPAI situation in the 2022-2023 season has been supplemented in the manuscript.
■line 70-80
Reviewer: Why do you mention to only H5N1 not H5Nx about infection to cats? What is the difference between H5N1 and H5Nx though the cases of the infection to cats with H5N6 have reported in China, 2014-16. As all researchers of AIV know, the origin of N1 on the early H5N1 virus differs completely from that on 2.3.4.4b H5N1.
In addition, are the cases in cats the relatively major case among “mammalian case with GsGd H5 viruses", aren't it, though the mammalian cases are rare?
And, mammalian infection with 2.3.4.4b HPAIV were more frequency than with other clade HPAIV, and the number of the registered strain(295) in GISAID is over half of total number(452) of HPAIV derived from mammals except human. The almost all mammalian strains in 2.3.4.4b were isolated in two to three years though HPAI outbreaks have occurred worldwide over 20 years. How do you think the relationship between this case in cat and these facts?
Our answer: Information on other mammalian infections caused by H5Nx has been added in the text.
There are several possible explanations for why clade 2.3.4.4b of avian influenza (AI) is frequently isolated in mammals:
- Adaptation: The clade 2.3.4.4b strains might possess genetic adaptations that make them more capable of infecting and replicating in mammalian hosts.
- Receptor Binding Specificity: The hemagglutinin (HA) protein of the clade 2.3.4.4b strains might have a receptor binding specificity that allows them to attach to receptors present on mammalian cells, facilitating infection.
- Transmission Dynamics: The clade 2.3.4.4b strains might have characteristics that enhance their transmission between birds and mammals or even among mammals.
- Ecological Factors: Environmental factors, including host populations and ecological interactions, might contribute to the prevalence of clade 2.3.4.4b in mammals.
- Undetected Infections: Clade 2.3.4.4b infections might be undetected in other species or might not cause severe symptoms, making them more likely to be identified in mammalian hosts.
According to Rabalski et al. the most possible answer is clade 2.3.4.4b with point mutations indicative of initial mammalian hosts adaptations. We have included this information along with the appropriate citations in the manuscript.
■line 244
Reviewer: You should discuss that there were lesions in the brain but RNA was not detected in that. Furthermore, the immunohistological results will make readers more understanding if there are.
Our answer: Thank you for drawing attention to the significant aspect of our study regarding the presence of histopathological changes in the brain, despite the absence of detected influenza virus RNA in this organ. Indeed, we identified changes such as necrosis and cellular infiltrates in the histopathological examinations of the brain, even though the RNA of the virus was not detected by the RT-PCR method. The possible explanations for this discrepancy is that the amount of the virus in the sample was below the detection threshold of our RT-PCR test. So the virus might have been present in such a small quantity that it wasn't detected in our test, but was still capable of inducing pathological changes. This appears quite feasible, given that the literature documents the identification of H5N1 within brain tissue. We have added such information in the discussion section.
■line 250-255
Reviewer: You should explain the pathogenicity of E. faecium and discuss its contribution to this dead case. As you said the responsibility of E. faecium for the emergence of the pathological lesions is impossible to determine, but you can discuss. Though E. faecium is known to the opportunistic pathogen in the field, it should be also explained in the manuscript.
Our answer: added
■line 256-283
Reviewer: This is the high-profile section in this manuscript. But we cannot understand which the risk of natural infection was high or low because the situation of HPAI in this season was not shown. You described that HPAIV in wild birds was lastly detected in Masovian Voivodeship on May 22, 2023. What about in the neighboring Voivodeship such as Lubelskie Voivodeship? The cat resided in the central or outlying in Masovian Voivodeship? Does cats usually catch wild waterfowl or small mammals such as mouses? In this version of manuscript, I receive the impression that the cat has been infected with HPAIV from the fed made of infected poultries not "natural" such as from wild birds. The possibility of the infection from "natural" should be more considered because the evidence of that from the fed or others were absent.
Our answer: We have added information that the cat originated from the central part of the Masovian Voivodeship; thus, a description of the situation in neighboring voivodeships seems unnecessary to us. We also emphasized more clearly in the abstract and discussion that we did not determine the source of infection. It was not our intention to stress that the infection occurred after consuming raw meat, as we do not know this; however, in the case of our cat, both possibilities were plausible.
Reviewer: Recently, the HPAI case in cat in Korea was reported to be caused by the fed made of infected poultries. (https://en.yna.co.kr/view/AEN20230802003000320) Thus, even now the HPAI infections by improper feds can be occurred. But your explanation is unfair because the reference you cited is old. You should be described that EU or Poland government takes measures to reduce the risk of HPAI spread after the incident you cited. For example, shipped meats from the farm must be recalled back to 21 days if HPAI occurred in the farm by EU regulation.
Our answer: We agree with the Reviewer’s comment. Surely the EU and Polish government takes measures to reduce the risk of HPAI spread. The relevant paragraph has been modified accordingly.

Reviewer 3 Report
The manuscript entitled “Fatal A/H5N1 avian influenza virus infection in a cat in Poland” reported a fatal case of A/H5N1 influenza virus infection in an outdoor cat. In general, A/H5N1 virus cross-species infection in mammalian animal host is of great significance to public health as such virus can further evolve and potentially spread to humans. However, this article mainly focuses on the description of disease progression, pathological examination, and virological and bacteriological testing results. It can be greatly improved if the H5N1 virus was isolated or sequenced to confirm the virus clade and molecular characteristics associated with high virulence.
Some major and minor comments:
1. Line 162-172: data from blood test and hematological analysis were reported here. Can authors comment on the significance of these changed numbers? Do these alterations suggest a particular disease?
2. Line 232: “this case report presents the first confirmed case of HPAI in a cat in Poland” is not an appropriate conclusion here. Authors cannot assume the H5N1 virus caused the cat death was an HPAI strain since there is no sequencing data to support it. Does their RT-qPCR detect only HPAI H5N1? There is no description on this.
3. How common is E.faecium infection in cats? Can it lead to lethal infection in cats? Please comment on this.
Author Response
Reviewer: The manuscript entitled “Fatal A/H5N1 avian influenza virus infection in a cat in Poland” reported a fatal case of A/H5N1 influenza virus infection in an outdoor cat. In general, A/H5N1 virus cross-species infection in mammalian animal host is of great significance to public health as such virus can further evolve and potentially spread to humans. However, this article mainly focuses on the description of disease progression, pathological examination, and virological and bacteriological testing results. It can be greatly improved if the H5N1 virus was isolated or sequenced to confirm the virus clade and molecular characteristics associated with high virulence.
Our answer: Thank you for your thoughtful feedback on our manuscript. We truly appreciate your suggestions regarding the isolation and sequencing of the H5N1 virus to better understand its clade and virulence markers.
Our primary objective with this publication was to document the disease progression, using ultrasonographic techniques to assess the severity of lung changes in animals with influenza, as is done in the human medicine, pathological examination, and testing results due to the rarity of such cases in Europe. In light of the significant spike in cases observed in Poland since June, we recognized an urgent need to equip veterinarians, cat owners, and veterinary services with detailed and valuable information. Given the prevailing sense of uncertainty and the challenges faced by many, it was of paramount importance for us to share the most comprehensively researched case from our region as expeditiously as possible.
During the review of our study, two papers focusing on the sequencing of cases of A/H5N1 influenza virus from Poland were published (our case was not included). All the examined cases belonged to the clade 2.3.4.4b of the influenza virus.
Once again, we value your input and hope that this clarification provides a better understanding of the intent and scope of our study.
Some major and minor comments:
Reviewer: Line 162-172: data from blood test and hematological analysis were reported here. Can authors comment on the significance of these changed numbers? Do these alterations suggest a particular disease?
Our answer: Thank you for drawing attention to the blood results of our cat with confirmed H5N1 influenza infection. The significant alterations observed in the biochemical parameters indeed warrant discussion.
The markedly elevated levels of aspartate aminotransferase (AST), alanine aminotransferase (ALT), creatine kinase (CK), and lactate dehydrogenase (LDH) primarily indicate substantial hepatocellular injury, as well as possible muscle damage. Such increases are suggestive of viral-induced cytopathic effects, which are consistent with what we might expect from an H5N1 infection, as the virus is known to have a broad cellular tropism. The elevation in total bilirubin further attests to hepatic dysfunction or injury. The hypokalemia, accompanied by a mild increase in sodium and chloride concentrations, suggests possible renal involvement or an electrolyte imbalance that could be secondary to the disease or other underlying conditions.
On the hematological front, the increased erythrocyte count coupled with a near upper-end normal hematocrit could indicate hemoconcentration, which might be attributed to dehydration or other secondary causes. The neutrophilia observed, with a corresponding decrease in lymphocyte percentage, is indicative of a systemic inflammatory response, commonly seen in infections.
While these alterations certainly suggest a severe systemic disease, they are not unique to H5N1 infection and can be seen in a variety of conditions, especially those involving liver, muscles, or systemic infections. It's important to interpret these results in conjunction with the clinical picture, history of exposure, and other diagnostic findings. In the context of a confirmed H5N1 infection, they provide a profound insight into the severity and systemic nature of the disease in this specific case.
We hope this sheds light on our findings and underscores the pathophysiological changes that H5N1 can induce in affected hosts.
We’ve added this information into Discussion part of Manuscript
Reviewer: Line 232: “this case report presents the first confirmed case of HPAI in a cat in Poland” is not an appropriate conclusion here. Authors cannot assume the H5N1 virus caused the cat death was an HPAI strain since there is no sequencing data to support it. Does their RT-qPCR detect only HPAI H5N1? There is no description on this.
Our answer: Thank you for highlighting the oversight on Line 232 and for pointing out the need for clarity regarding our RT-qPCR methodology. The multiplex RT-qPCR reaction described by Stefańska et al. allows for the simultaneous detection and differentiation of the N1 and N2 genes as well as the H1, H3 and H5 genes of influenza A viruses. These simultaneous three duplex real-time PCR assays make it possible to determine whether we are dealing with the presence of A/H5N1 RNA. We acknowledge and concur with your observation. Our PCR method was indeed not tailored to differentiate between HPAI and LPAI H5N1 strains. In light of this, we have removed the mentioned statement from our manuscript to ensure accuracy and avoid any potential misconceptions.
Reviewer: How common is E. faecium infection in cats? Can it lead to lethal infection in cats? Please comment on this.
Our answer: Thank you for raising the pertinent question regarding E. faecium infection in cats. E. faecium is a bacterium that naturally colonizes the digestive tract of cats. Beyond its natural occurrence, it's worth noting that E. faecium is also employed as a probiotic due to its potential health benefits. However, a growing concern within the veterinary and medical communities is the escalation of drug resistance exhibited by these bacteria, particularly resistance to antibiotics like ampicillin and vancomycin. It has been observed that such resistant strains of E. faecium can be isolated from the gastrointestinal tract of even healthy cats. Practices such as feeding cats raw poultry meat have been linked to the facilitation of this resistance.
While E. faecium is prevalent in the gastrointestinal system, systemic infections in cats attributed to this bacterium are relatively uncommon, mirroring the pattern seen with many opportunistic bacteria. When a systemic infection does occur, especially if it involves the central nervous system, the prognosis tends to be grim. Such infections often manifest in the backdrop of an immunocompromised state, attributable to factors like aging, chronic diseases, therapeutic interventions, or prior infections with another pathogen. In this specific case, we suspect the latter scenario.
We deemed it essential to elaborate on the co-infection with E. faecium for a couple of primary reasons:
- The strain identified was multidrug-resistant, rendering first-line antibacterial treatments ineffective.
- The concurrent presence of faecium complicated the clinical manifestations of influenza in this particular patient, making the diagnostic and therapeutic journey more intricate.
We’ve added this information into Discussion part of Manuscript.
Reviewer 4 Report
The manuscript is the detailed and meticulously executed case report of cat, that dead some days after onset of symptoms with clinical presentation typical of the A/H5N1 virus infection in domestic cats. Post-mortem molecular investigation detected the RNA of the A/H5N1 influenza virus in multiple organs. Also Enterococcus faecium was detected in almost all tested organs that may point to secondary septicemia, a common complication of many generalized viral infections. The possible source of the infection for the investigated cat is discussed.
The manuscript is well written and can be accepted in present form.
Author Response
Reviewer: The manuscript is the detailed and meticulously executed case report of cat, that dead some days after onset of symptoms with clinical presentation typical of the A/H5N1 virus infection in domestic cats. Post-mortem molecular investigation detected the RNA of the A/H5N1 influenza virus in multiple organs. Also E. faecium was detected in almost all tested organs that may point to secondary septicemia, a common complication of many generalized viral infections. The possible source of the infection for the investigated cat is discussed.
The manuscript is well written and can be accepted in present form.
Our answer:
Dear Reviewer,
Thank you for taking the time to review our manuscript. We deeply appreciate your detailed assessment and are pleased to hear that you found the case report to be meticulously executed. We are encouraged by your positive feedback and the acknowledgment of the relevance of our findings. Your recognition of our work and the possibility of accepting it in its current form is highly motivating.
Warm regards

Round 2
Reviewer 2 Report
This report is important to diagnosis for HPAI by clinical veterinarians. Virus detection, histpathology results and course of treatment will be helpful for HPAI case in cat which will happen in the future. These seem to be well described in the revised manuscript. However, the authors did not describe enough the HPAI situation, though the situation of HPAI in birds also will be important and helpful information to suspect HPAI case in cat.
The revised version seems to be well improved, but the authors didn't correspond enough to some comments.
■Required
The authors answered "Surely the EU and Polish government takes measures to reduce the risk of HPAI spread. The relevant paragraph has been modified accordingly". But, I couldn't find the revised or added sentences in the revised manuscript.
■line 71
The authors only added the references. As I said in the first comment, the situation of HPAI in 2022-2023 season is important more than that in 2021-2022 season. The authors should have described the situation of HPAI in Poland or EU in 2022-2023 more detail than 2021-2022.
I understand that the HPAI situation in 2022-2023 has not summerized yet enough in web or reports. However, at least the number of HPAI outbreaks in Poland or EU in 20222-2023, or whether that is high or low or comparable to that in 2021-2022 should be described in the manuscript.
■line 330
The authors answered "a description of the situation in neighboring voivodeships seems unnecessary to us.". But, the authors described "as wild birds are migratory, and may carry undetected, subclinical/mild infections" in the same section. If so, a description of the situation in neighboring voivodeships seems necessary. Or, the author should show the basis (e.g. the distance to neighboring voivodeships) of the decision that the risk from wild bird is low.
Author Response
Dear Reviewer,
Thank you once more for your attempts to improve our manuscript.
Below our corrections:
Reviewer: The authors answered "Surely the EU and Polish government takes measures to reduce the risk of HPAI spread. The relevant paragraph has been modified accordingly". But, I couldn't find the revised or added sentences in the revised manuscript.”
Our answer: The section has been added (lines 439-457); it was inadvertently omitted in numerous previous revisions.
Reviewer: The authors only added the references. As I said in the first comment, the situation of HPAI in 2022-2023 season is important more than that in 2021-2022 season. The authors should have described the situation of HPAI in Poland or EU in 2022-2023 more detail than 2021-2022. I understand that the HPAI situation in 2022-2023 has not summerized yet enough in web or reports. However, at least the number of HPAI outbreaks in Poland or EU in 20222-2023, or whether that is high or low or comparable to that in 2021-2022 should be described in the manuscript.”
Our answer: Corrected. Now it is inserted (lines 70-78)
Reviewer: “The authors answered "a description of the situation in neighboring voivodeships seems unnecessary to us.". But, the authors described "as wild birds are migratory, and may carry undetected, subclinical/mild infections" in the same section. If so, a description of the situation in neighboring voivodeships seems necessary. Or, the author should show the basis (e.g. the distance to neighboring voivodeships) of the decision that the risk from wild bird is low.”
Our answer: We agree that “unnecessary” was an incorrect expression, thank you. We reformulated this fragment and added more details (lines 368-373)
